# Cain and Abel: Re-Imagining the Immigration 'Crisis'

**Abi Doukhan**

Department of Philosophy, Queens College, Flushing, NY 11367, USA; doukhana@yahoo.com

**Abstract:** This essay proposes to interpret the significance of the so-called immigration crisis in the light of the ancient story of Cain and Abel. Much more than a mere conflict between brothers, this essay will argue that the story of Cain and Abel presents two archetypal ways of dwelling in the world: the sedentary and the nomadic. As such, the story sheds a shocking new light on our present crisis, deeply problematizing the sedentary and revealing in an amazing *tour de force*, the hidden potentialities of the nomadic and the powerful rejuvenating force that comes with its inclusion and welcoming in the sedentary landscape that characterizes our Western societies.

**Keywords:** Christian; Jewish; Biblical; Cain; Abel; immigration; ethics

## Introduction

Listening to the current debate on immigration, I am surprised to see how some of my Christian and Jewish counterparts seem completely fine with some of the more rigid stances on immigration, including the deportation of illegal immigrants, the separation of immigrant families, and the suspicion shown towards immigrants aspiring to set foot on American soil.[1] This hardened stance towards immigration on the part of those claiming to follow the message of the scriptures is surprising inasmuch as the Bible seems to shed a positive light on not only the exiled, but the very condition of exile. From the exilic calling of Abraham and the exile of the Hebrews from Egypt to the Christian calling to be "strangers in the world", the Bible seems sympathetic to the condition of exile. Moreover, the Bible is replete with injunctions to love the stranger and to care for him/her. This is not only one of the central themes of the Hebrew Bible, but is also evident in Christ's behavior towards the marginalized and the despised of his time—the prostitutes, the tax collectors, the gentiles, women, etc. Contrary to the common sense of some of our contemporaries who see the exiled with suspicion and distrust, the Bible not only sheds a positive light on the condition of exile, but also instructs us to love the exiled.

The question is, however, why the Bible places such an important emphasis on the condition of exile as well as on the need to welcome the exiled. Why is exile seen in such a positive light? More importantly, why does the Bible teach us to care for the exiled? How is this an essential duty as a Christian or as a Jew? This essay proposes to address these questions from the perspective of a very short story narrated in the Hebrew Bible: the story of Cain and Abel. Now, the choice of this particular story will appear to some to be somewhat peculiar. It is difficult to see the connection between that particular story, which takes place between two brothers, and the situation we are in of choosing whether we should welcome or not the exiled among us. What we forget though is that this story

---

depicts far more than a mere squabble between brothers. The story of Cain and Abel has archetypal value and, as such, constitutes a story that illuminates something about the human condition at large and, more specifically, about the way that we relate to the other in our world. The story thus functions as a mirror to our present condition and can give us a deeper understanding of how we are to relate to the strangers in our lives, as well as to the divine intent for the exiled and the strangers that inadvertently enter our world. The story of Cain and Abel thus arguably holds a profound lesson with regards to the immigration problem, and it is now to this story that I would like to turn.

1 See the following Washington Post article on the hardening of Evangelical stances on immigration: https://www.washingtonpost.com/news/acts-of-faith/wp/2018/06/18/why-many-white-evangelical-christians-are-not-protesting-family-separations-on-the-u-s-border/?utm_term=.2866dec91c96. When it comes to the Jewish stance on immigration, although it has historically been pro-immigration and has supported protecting immigrants' rights, the Orthodox Jewish vote in the last few years has shifted to privileging Israel over issues such as immigration, social justice, etc. https://www.jpost.com/Diaspora/I-think-its-Israel-How-Orthodox-Jews-became-Republicans-616442.

The story of Cain is an intriguing one, riddled with enigmatic allusions, twists, and turns. From the beginning, he is given a central position in the world, called "a man" by his mother upon her giving birth to him, and ascribed an almost divine character. Already in the story of his birth, we have a sense of his importance, of his centrality in the world.[2] When Eve gives birth to Cain she exclaims, "With the help of the Lord I have brought forth a man" (Gen 4:1), whereas the birth of Abel is only mentioned in passing. Moreover, as his name and profession as a tiller of the soil indicate, he is also profoundly grounded in the world—at home in it and in full possession of it. The root for the name Cain, *qanah*, meaning "to acquire", alludes to the possessive and masterful stance of Cain. Thus, the central and masterful stance exercised by Cain later on is already inscribed in his very name. He is born under the sign of mastery, of acquisition. Cain's destiny will be marked by the desire and ability to possess, to acquire, thereby ensuring the centrality and strength of his stance in the world. In other words, Cain's central and possessive stance in the world is that of the hard-working success story that has carved out a place for itself in the world! It is the very epitome of the American dream! It is what all of us are aspiring to become! Cain is what we *have* become—the proud owners of our success and of our territory.

Next to Cain, Abel, seems almost insubstantial. His name means "vapor" or vanity, announcing a personality not intent on possessivity or mastery. His name means "fleeting". Incidentally, the name Abel will come up again in the book of Ecclesiastes to speak to the ephemeral character of life, of the fact that nothing lasts forever, nothing endures. Likewise, Abel, as his name and as his destiny will indicate, has no substance, no hold on the earth, and no chance is given to him to make a mark on the latter. Moreover, Abel's profession further alienates him from the land: he is a shepherd, condemned to perpetual movement, to perpetual exile, ever searching for fertile land for his flocks, always on the move! As such, he seems to be the perfect parasite, condemned to wander lands owned and cared for by others, ever reaping benefits from those who have worked hard, living on borrowed land, and enjoying unearned advantages. Thus, from the onset of the narrative, Abel is on the margins of history, having built and accomplished nothing! He does not own and has not tilled the land, but here he is, asking to be admitted upon it for the pasturing of his flocks. Abel is thus the one who has failed to become important, the one who has failed to make a name for himself; he is the wandering Jew, the immigrant, the refugee, the political exile.

Yet, in our story, it is Abel whom God chooses to acknowledge; it is his offering that God welcomes, whereas Cain's offering goes largely ignored. Now, this is interesting! It is as though the Biblical narrative seems to distrust Cain and his central possessive stance on the world. Contrary to traditional protestant ethics where wealth and material success is seen as a sign of divine election,[3] our Biblical narrative seems to find these lacking. Cain, in his comfortable stance in the world, is not seen favorably by God. He is missing something! The subjectivity at home in the universe, the hard-working homeowner, who has earned his bread at the sweat of his brow, is not seen as a success story in our Biblical narrative. To own a patch of land is not enough and does not point to divine favor. Far to the contrary, the self-sufficient landowner represented here by Cain is seen with incredible distrust. In fact, he is largely ignored by the divine gaze, which prefers to consider his brother Abel. But what does the

---

2    André Lacocque also makes this observation: "Cain occupies center-stage all along. Abel is narratively (as well as ideologically) eclipsed by his brother" (Lacocque 2010, p. 41).

3    Cf. (Weber [1905] 1930).

latter have that the former does not? Precisely this: he has nothing. Abel, whose name means "vapor", "breath", is a migrant on the earth. He is a shepherd, which Biblically speaking means that he has no claim on the land. He wanders on a land that does not belong to him. He is in perpetual exile, on borrowed territory, dependent on landowners for his living. He is the refugee, the immigrant among us. Yet, it is to him that God turns his gaze as though it were precisely his condition of exile that God finds attractive.

But why is that? Why is exile more pleasing to God than the sedentary condition? Why is God so seemingly unjust to the homeowner at home in the universe, privileging his exiled, immigrant brother? Our text does not give us any clear-cut reasons as to why God chooses to ignore Cain. However, what we know for sure is that this act of disrespect on the part of God profoundly alters Cain's stance in the world. Indeed, the text says that Cain's face "fell". This is significant when one realizes that the face constitutes more than a mere part of the body but, rather, symbolizes the self's dignity and personhood. God's actions have the result of destroying in Cain what constituted his dignity—his manhood and humanity. It is his own deposition, his own death, that Cain sees in Abel's individuation by God. Likewise, we also, like Cain, feel a certain discomfort at the presence of an other, a stranger, in our hereto homogenous, "safe", and familiar neighborhoods. The other, the stranger, the immigrant, like Abel unto Cain, is indeed a threat to our comfortable stance in the world, to our hard-earned place in the sun; he is also a threat to our vision of the world, to our values and ways of life. So, to share the world with this good-for-nothing intruder does not seem to be in our job description as humans. This is not, however, the take of the Biblical story where God seems to despise Cain over Abel. The question of course is why. Is there a deeper intention behind God's seemingly unjust actions?

One wonders if there is not perhaps meaning to be drawn from God's actions towards Cain. Perhaps, there is a pedagogical intention behind this pain inflicted by God upon Cain. But to understanding this intention, we must go back to what constitutes Cain's problem. I would argue that the sacrifice of Cain does not contain the key to the nature of his sin. It is the passages prior to the event of the sacrifice that give indication to Cain's problem. Cain's problem is not so much in his intentions or in his actions as in his general stance in the world: a central stance, which, as such, remains essentially oblivious to an other. Cain's problem lies then not so much in his performing the wrong rite or in not being attuned to the spiritual realm as it lies in a lack of concept of otherness.[4] It is then not that Cain is not a good person or even a good "Christian". Certainly, he is to be admired as a hard-working individual, who has earned his place in the sun. He is also, to be sure, an engaged believer because he is the one who comes up with the idea of sacrificing to God. However, Cain's problem is not a spiritual one; it is an ethical one. Inasmuch as he has no concept of ethics, he likewise has a poor concept of transcendence and of the spiritual realm, for to lack a concept of the other, to lack sensitivity to the other, is ultimately to lack interest in God as the great Other.[5] The God of Cain is a God to his measure—someone he thinks he can impress or manipulate. As long as Cain does not see Abel, one might argue that he does not really see God. This is evident in the way that he ultimately totally misses the mark in his sacrifice.

This is where God's way with Cain becomes interesting. Indeed, what better way to open Cain up to the dimension of the other than through the experience of suffering or pain? Inasmuch as pain constitutes the disturbance of a self's complacent and comfortable stance in the world, it has an ethical

---

4　André Lacocque goes as far as to diagnose Cain as a narcissist: "For Cain's profile would remain incomplete without diagnosing him as narcissistic. Only a narcissistic blindness can being someone to 'even gracefully kill a brother since he is not 'me,' therefore less sacrosanct, 'less human' than one is oneself'" (André Lacocque, *Onslaught of Innocence*, p. 103. Lacocque is here quoting (Becker 1962, p. 162).

5　Emmanuel Levinas comments on this beautifully when he describes ethics as the way to God, referred in this passage interchangeably as the "Enigma" or the "Infinite": "Morality is the Enigma's way . . . The I approaches the Infinite by going generously toward the You . . . I approach the Infinite insofar as I forget myself for my neighbor who looks at me; I forget myself only in braking the undecipherable simultaneity of representation, in existing beyond my death. I approach the infinite by sacrificing myself" (Levinas 1996, p. 76).

significance. Pain interrupts the hereto self-sufficient, self-absorbed, and self-reliant stance of the hard-working Cain. However, this pain is not meaningless or absurd. There is a meaning behind this painful experience. The pedagogy of pain is a pedagogy of otherness. It is the personhood of his brother Abel that is signified behind Cain's pain, which arises from Cain's painful experience of rejection. In respecting Abel and *not* Cain, God allows for Abel to rise up, for the first time, as a person in the realm of Cain.[6] For the first time, Cain takes notice of his brother; for the first time, he sees him and notices his presence in the world. For the first time, Cain realizes that he is not alone in the world, that he is not the center of the world. This unraveling of Cain's hereto central and masterful stance, this expulsion on the part of God, is, thus, not performed in order to destroy him. This is not a case of arbitrary favoritism. Behind God's preference of Abel, it is Cain that is sought out. In the turning of his face towards Abel, God is seeking Cain. The whole event aims at saving Cain from an existence from which all otherness is blotted out. The pain inflicted upon Cain constitutes his first experience of otherness and, as such, an opportunity for him to arise to true self-hood and to true individuation as a consciousness no more self-enclosed but genuinely transcending itself towards an other. The pain is there then not to destroy Cain but to give him his first experience, his first lesson of otherness. In other words, far from disintegrating the self, the painful experience of the other frees the self, releases it from its "enchainment to itself", from its suffocation.[7]

God is then not so much trying to annihilate Cain as to release him from the prison of his ego. God is not so much trying to destroy Cain's world, as to broaden it to include the dimension of the other, to make it into a shared world. It is then not the destruction of Cain that is aimed for by God's pedagogy of pain, but his elevation to true selfhood.[8] What makes for the self's true dignity is not material success, achievements, or even hard work, but a certain sensibility to otherness. The elevated self is not the *successful* self but the *ethical* self. True selfhood is not that of a central, hard-working self, who has carved for itself a place in the world, but rather that of a sensitive, vulnerable self that has awakened to the dimension of the other.[9] The pain that Cain is experiencing as the end of him is in fact the opening up of the possibility of otherness. Such is, then, the pedagogy of pain: to open up the self to a dimension other than itself, beyond itself, otherwise than being and, as such, to allow for the genuine self-transcendence necessary to true worship. For only a self that has a concept of the other can genuinely address God, can genuinely access transcendence. It is ethics that, in the case

---

6　André Lacocque accurately speaks of a "resurrection" of Abel through the divine glance: "Abel . . . died 'before his time,' the first human to have his life cut short by someone else; and, in the image of his existence, Abel's death is without luster. Or so it would be for the astonishing divine preference for the living Abel's grand gesture of his offering, and for the dead Abel's word. Then, the transformation is total. Abel is resurrected. Or perhaps should we say that he is quickened for the first time" (André Lacocque, *Onslaught of Innocence*, p. 65).

7　Emmanuel Levinas has developed at length the idea that true freedom does not consist in doing what one wants, but rather in opening oneself up to the infinite dimension of the other. To do so, although setting limits on self- interest, would entail freeing oneself from one's selfish and narrow preoccupation with oneself: "The other absolutely other—the Other—does not limit the freedom of the same; calling I to responsibility, it founds it and justifies it" (Levinas 2004, p. 197).

8　André Lacocque also observes this pedagogical concern on the part of God in his observation that God does not kill Cain in retaliation for his brother's murder but, rather, merely exiles him: "Incidentally, Cain is not slaughtered . . . The culprit's punishment is strikingly not revenge. It is for fairness sake; it responds to Abel's blood clamoring for justice . . . The spilled blood of Abel crying for justice . . . does echo another shout, although an oxymoronic silent one. For Cain's is 'an act of murder screaming for help, a murder screaming for recognition' to borrow the words of Michael Eignen. God's address to murderous Cain shows that both screams are heeded here and now. As re-readers of the tale, we know that the latter ends on a different, hopeful note: 'people began to call on the Name of Yhwh.' But the divine discourse to the 'fallen' human precedes this appeal to God" (André Lacocque, *Onslaught of Innocence*, pp. 16–17. Lacocque is here quoting Eigen 2002, p. 109).

9　We see this at length in Emmanuel Levinas' philosophy where the mature self is defined as responsible over and against the immature/innocent/unjust self defined as enjoyment (cf., *Totality and Infinity*). This idea is further developed in the philosophy of Augustine Shutte in his exploration of the Bantu concept of Ubuntu: "If one takes the personal community achieved by personal growth as the ultimate standard for ethics then one is able to overcome the apparent conflict between self-love and love of others that other approaches to ethics find so difficult to deal with. Genuine self-love turns out to be love of precisely that in myself that I most deeply share with others, our humanity. This is the attitude of UBUNTU. So *eros*, the desire for personal fulfillment, and *agape*, desiring the fulfilment of others for their own sake, are seen as two sides of the same coin rather than as enemies" (Shutte 2001, p. 66).

of Cain, must usher in metaphysics. Without a sense of ethics, no metaphysics, no spirituality, no religion is possible. It is then because of Cain's obliviousness to the other that he could not attain true worship, true self-transcendence towards God as transcendent and other, and that his sacrifice was not respected. Only upon seeing the face of his brother, Abel, would Cain rise above nature and attain the spiritual dimension of his very being to which he hereto was oblivious. Only upon learning to share the world with his brother, Abel, would that world be genuinely sanctified by the pious actions of a wholly lived sacrifice!

We are now in a position to finally understand the hidden blessing presented by the exiles, refugees, and immigrants that burst into our lives. The exiles and immigrants among us have a very special lesson to teach us—a lesson of transcendence serving to awaken the self to a dimension beyond itself, to more than itself, to an other. The story of Cain and Abel is then the best illustration of the higher calling contained within the encounter with the exiles and immigrants among us. It signifies an experience, an encounter with otherness, and as such, with transcendence. Affectivity and sensibility to otherness are thus awakened and heightened in the self by the trauma and suffering associated with the encounter with the exile. As such, spiritual perceptions are heightened. Having shown ourselves capable of welcoming a human other as other, we are now ready to engage with a God who is himself radically other, ever disturbing, ever challenging the ego's plans and projects for itself! The temptation of idolatry—of worshipping a God in the image of the self's delusions and fantasies—can only be overcome like this. The ability to welcome a stranger in his difference, in his disturbance, shows a deeper ability to overcome idolatry in the spiritual realm—that is to say, a readiness to be disturbed, to be overwhelmed, to be taught by a God who is himself the ultimate immigrant, a stranger in the world.

There can thus be no genuine piety, no genuine "Christian nation" without a fundamental concern for otherness, for the immigrant and refugee at our door. However, there is more. The welcoming stance towards the other, as uncomfortable as it is, constitutes the condition for the emergence of the mature human self. Had Cain welcomed Abel, he would have emerged in his true human stature and overcome the stunted self-centered stance that hereto characterized him. What makes us fully human is not the ability to cultivate the ground—that a machine can do just as well—but rather the ability to cultivate relationships. We are not, as Aristotle thought, *rational* animals, but rather, *relational* animals. Our highest self emerges in relation, not in rationality, as was conceived by Western philosophical thought. This is perhaps what we might learn as a nation with immigrants gathering at our borders. What makes us great as a nation is not our technological prowess, our scientific advancements, or even our ability to build wealth for ourselves. A nation of robots can do the same. What makes a nation truly great, what makes it in fact a civilization, is its capacity to rise above mere survival mode and open onto the spiritual and human dimension of the other. This refinement of a nation into a civilization is only possible, however, through the oft painful and uncomfortable act of sharing the world. This is the condition of any attempt at building a genuine civilization. Without this ability to reach out in an inclusive gesture to humanity, we are condemned like Cain to remain stunted in our growth and maturation as a community. Thus, the welcoming of the stranger, far from limiting us, in fact constitutes the prerequisite of our full flourishing as a civilized world. The other, far from stunting our growth and prosperity as a nation, in fact serves to bring this growth and flourishing about.

Having said this, a number of important objections arise. The main one being that the Biblical lessons on the welcoming of otherness are all very good, but they are not realistic. That is to say, the Biblical injunction that the stranger is to be always welcomed is neither feasible nor recommended. There are three objections that arise when one thinks of this teaching on hospitality towards the stranger: (1) the immigrant presents a potential threat—we do not know who he/she is, and therefore, it is important to create a filter system at the border that only lets the good immigrants in and keeps

the "bad hombres" out;[10] (2) as a country, we cannot possibly begin to welcome all of the refugees who knock on our door because we simply do not have the resources to take care of all of these people; (3) admitting too many immigrants from other cultures and religions jeopardizes national identity—we will eventually lose our culture, religion, and values with the incoming of too great a flux of immigrants. Interestingly, our story contains the elements of a response to each one of these objections. So, it is to these that I would like to now turn.

Let us begin with the notion of the potential threat posed by the immigrant and the consequently wise measures to create a "filter system" for immigrants. It is fascinating to me that, when we read the story of Cain and Abel, our text reverses the whole question of whether the stranger poses a threat and how we are to mitigate this threat. In our text, it is not Abel who poses the real threat but Cain! It is Cain who, in the end, kills Abel and not the other way around. The question of course is how we, as the children of Cain—owners and workers of the land—might harbor, unbeknownst to ourselves, the seeds of murder. What is it, in our very positioning as the central characters of the story, as those who have "acquired" land, success, and wealth at the sweat of our brow, that might hide a motif for murder? To uncover these seeds, we must explore our own fear of the stranger. The fear we have is that some immigrants, because they have not been properly filtered at the border, will present a violent threat to us. I am not yet talking about the threat to culture here inasmuch as this will be the third point that I will discuss. I am simply talking about a genuine existing fear that the strangers that we allow in would turn out to be a threat to the lives and dignity of our own.[11] This fear was perfectly summarized in the US president's words about Mexican immigrants: "They're sending people that have lots of problems, and they're bringing those problems with us. They're bringing drugs. They're bringing crime. They're rapists. And some, I assume, are good people".[12]

We may laugh at this almost caricatural depiction of immigrants, but the fear of the stranger turning out to be a rapist or a murderer is a genuine one among many of us. The question I would like to pose here however is not whether this fear is founded or not. There is enough factual and statistical evidence that it is not a big problem. Yes, immigrants sometimes pose a threat, but not much more—in fact, much less—than our own do.[13] I would argue that a greater danger looms than that of the occasional threat posed by the immigrant: that of becoming ourselves a threat and a menace to the lives and dignity of others. Becoming ourselves the perpetrators of a crime against humanity is a much more urgent and pervasive problem than that posed by the isolated crimes committed by a handful of immigrants. But what is this danger that we pose to others?

It can be argued that the very attitude by which we come to see the other as potentially threatening is itself a form of deep and pervasive violence. Each time we choose to reduce, limit, stunt, and curb the other into our own representations and projections, we are committing an act of violence towards them. This is not a physical act of violence, but it is a moral act of violence, by which we strip the other of their personhood by objectifying them into a threat. At that moment, they are no more a person, worthy of being known, discovered, encountered, but the rigid and fixed object of our fears and of the projection of those fears. They are no more a living and breathing human being with infinite possibilities and aspirations, but the reified constructs of our perverse and paranoid imaginaries. To use Buberian terminology, they are no more perceived as a human "thou" but rather are reduced to a

---

[10] In her book *Adios America*, Ann Coulter proposes filtering immigrants like the National Football League (NFL) selects good from bad players (Coulter 2015). The problem with this argument is the question of who determines what "good" and "bad" means. Ann Coulter suggests that if an immigrant is crippled or blind, he/she should not be admitted into the country, chillingly reminiscent of Nazi policy regarding people with special needs.

[11] One can see evidence of this fear in Ann Coulter's *Adios America*, in which she takes inventory over several chapters of news stories featuring immigrants exhibiting various forms of criminal behaviors from murder to sex trafficking.

[12] Speech made on 16 June 2015.

[13] Cf. https://www.cato.org/blog/illegal-immigrants-crime-assessing-evidence.

mere object, or an "it".[14] As such, our distrustful projections on the unknown other embodied by the immigrant result in our stripping them of their very breath, life, and humanity.

However, there is more. History shows how this attitude always eventually translates into physical violence and even murder. The reification of the stranger into a threatening object is only the beginning. Doing so provides both the moral and rational grounds for physical violence against the stranger, leading at times to the very extermination of that other. That the stranger is a threat thus provides the rational ground for exerting violence against that other and even for killing them. That the stranger is less than human, degraded to the objective categories of "rapist" or "murderer", provides the moral justification for their extermination. While it is not morally acceptable to kill a human being, it becomes easier to do so once that human being has become reduced to a mere concept or category, especially if that concept has a negative connotation such as "murderer", "rapist", or even "invaders", "illegal immigrants", "aliens", etc. History has sadly shown this to be true, from the Nazi genocide to the Rwandan one. Such genocides would not have been possible if the other had not been first dehumanized and degraded to the level of subhuman, even bacterial categories. The Jew first had to be assimilated to "vermin" and the Tutsi ethnic group to "cockroaches" before it became morally justifiable to kill them. In light of these historical facts, the present demonization of immigrants as "troublemakers", "rapists", or "murderers" becomes profoundly problematic.

However, inasmuch as we stand in the line of Cain, there is yet another way that we as a successful and resourceful community might pose a threat to the strangers at our doors. This has everything to do with our stance, like Cain, as the acquisitors and possessors of the land. One might wonder, at first, how this might be a problem. We are, after all, the ones who have worked and tilled this land. Are we not, as such, entitled to the fruit of our labor? Why would we not be the possessors of the land? In particular, why should we share it and the fruit thereof with someone who has not lifted a finger so far to work this land? Why should strangers benefit from our work and eat the fruits of our labor? We worked hard to make our country great, why should our success be now sliced up and shared with others? The issue here is not even the common misconception of the stranger as a sort of "leech", ready to pounce on the fruits of my labor. It is factual knowledge that immigrants do not move to another country in order to passively receive its benefits, but to work.[15] The issue here has to do with confusing ownership of the *land* and ownership of the *fruits* of the land.

While we are certainly entitled to the fruits of our labor on the land, we are not entitled to the land. While we may possess the houses we build, the produce that we harvest, and the animals and technology we have purchased, we ought never to think of ourselves as possessing the land itself. In the Hebrew Bible, the land, like water and oxygen, can never become the sole possession of a given person or even group. It belongs to God.[16] While we have a right to the fruit of our labor on the land, we have no natural right to the land, and this is precisely because we did not *labor* for the land or the waters or the air, but rather *received* them as part of our *common* inheritance as human beings. Although this notion might ring strange to our Western ears—we have become so used to violently taking and possessing whatever our eyes desire—it is not completely foreign to us. After all, our constitution does mention that "no person shall be deprived of life, liberty and property". Yet, I am always shocked at how these founding words of our democracy have been systematically used by the few to deprive the many.[17] Instead of hearing these words to mean that *all* human beings have a natural right to

---

14    Cf. (Buber [1923] 2010).
15    This is again Coulter's argument. However, research has shown that illegal immigrants actually do not receive welfare benefits, only legal immigrants, who, similar to most Americans doing low-paying jobs, need welfare to make ends meet. https://www.huffingtonpost.com/entry/no-undocumented-immigrants-arent-stealing-your-benefits_us_5a144263e4b010527d6780b0.
16    Leviticus 25:23.
17    Howard Zinn gives an analysis of this in his formidable *A People's History of the United States*, arguing that "governments, including the government of the United States—are not neutral, that they represent the dominant economic interests, and that their constitutions are intended to serve these interests". He goes on to observe that "the Constitution omitted the phrase "life, liberty and the pursuit of happiness", which appeared in the Declaration of Independence, and substituted "life, liberty and property" in order to ensure that the one-third of Americans who actually owned property would be guaranteed

unrestrained access to land, we have made it into a way for the haves to consolidate their hold on their own property, even to the detriment of those in need. Let us not forget the Native American genocides and forceful landgrabs that constitute the deeply problematic foundations of our nation and of its seemingly inclusive Constitution, notwithstanding the marginalized groups dwelling in our midst that both historically and in the present time have never really seemed to fit under the category of full-fledged persons entitled to life, liberty, and property.

Perhaps we need to remember that there are things that we are not entitled to, that there are things in this world that we may not possess. I would venture to predict that realizing this might profoundly shift the way that we approach the immigration "problem". It might profoundly shift where and who the problem really is. Our text teaches that it is *we* who are at the root of the immigration "crisis" and not the strangers at our door. *We* are the ones who, through our unjustified encroaching upon and hoarding of territory, are causing this crisis and this problem. Were we to accept that the land is and was never ours but was always meant to be shared, we would have no need for tighter border controls and for building walls. We would simply accept immigration as a natural fact of life—which it is—and receive the other as having claims to the land as legitimate as ours. Of course, I am not suggesting that the fruit of our labor, our houses, our possessions are all to belong now to the stranger. These are legitimately ours because we have worked for them. However, the land was never ours to begin with, and we need to stop acting like we have the sole title to it. Although realizing this would not make things any simpler, it would radically transform our attitude towards and the measures taken when faced with the stranger at our door in a way that, arguably, would be much more humane, ethical, and, might I say, fair.

However, this leads us directly to our second objection: What if we *did* open our borders in a welcoming stance to the stranger? What would happen then? Would the powerful influx of immigrants not jeopardize our resources, which, unlike our stance of infinite hospitality, are themselves finite? If welcoming the stranger at our door is the right thing to do, is it the practical thing to do? Again, our story offers us some interesting insights on this issue. Moreover, again, our story reverses the issue completely. In our text, it is not Abel who poses a threat to the resources available, but Cain. It is Cain who, upon killing his brother, is told by God that "the voice of your brother's blood is crying to me from the ground. And now you are cursed from the ground which has opened its mouth to receive your brother's blood. When you work the ground, it shall not longer yield to you its strength" (Gen. 4: 10–12). In this passage, there is a direct correlation between Cain's murder of his brother and the ground's losing its productivity or its "strength". Now, this is very interesting and worth exploring. It is not immediately clear how the two are related. How is it that a murder between two people can have such an impact on the land and its productivity?

In order to understand this, we must more deeply explore the specific psychology of murder. For murder does not just happen, there is a certain psychological terrain that must first exist in order to prepare the ground for murder. In the case of Cain, the psychological terrain is that of a man who has never been given a sense of his limits. Let us remember the story of his birth. Upon giving birth to Cain, Eve exclaims that she has "gotten a man with the help of the Lord" (Gen. 4: 1), after which Abel is born, almost in passing. Clearly, he is not the main character, Cain is. The latter will further consolidate his power in the world by acquiring land (as his name indicates: Cain is derived from the Hebrew root *qanah*, which means to acquire). The sole owner of the land and of its fruits, nothing is now impossible to him. He is all-powerful and without limits, until, of course, the event of the sacrifice and God's acceptance of Abel over Cain. We know now that this event was no arbitrary occurrence but constituted an attempt for God to awaken Cain to his limits, to the fact that the world did not solely belong to him but also to his brother, that the world that he had worked so hard to acquire for himself

---

their rights by the government over their already existing property and NOT that all Americans, that is to say, the remaining two-thirds of Americans, had an entitlement to property in the United States of America (Zinn [1980] 1994, pp. 98–99).

was in fact a shared world. However, by the time God gets to Cain, it is already too late. Unable to accept his limits and the fact that not everything lies within his power and grasp, Cain kills Abel in a desperate attempt to regain control over the world.

So, the terrain for murder is really a sense of entitlement to everything that exists as well as a lack of awareness of one's limits when it comes to those resources. We are now in a better position to explore the connection between murder and the degradation of the land and of its fruits. We have seen that Cain's problem lies in an inability to set limits upon his greed for acquisition of the land in order to share it with another. This inability results in a subjectivity, which has a limitless sense of its power. The other has never taught it to curb its power and to limit its greed. Beyond the obvious ethical problem that this attitude poses, there is however also a practical/material problem. A subjectivity that has never been taught to set limits upon itself with regards to a human other will engage with the land in the same way, leading to the overuse and overexploitation of natural resources and to their eventual depletion. A limitless subjectivity thus poses a threat not only to the human others that inhabit its land—by depriving them of access to this land—but to that land itself![18]

Thus, according to our text, it is clearly not the stranger at the door who poses the real threat to natural resources, but a subjectivity or psychology unaware of its own limits and therefore unable to curb its own greed and instinct of possession when working the land. It is not a given influx of people, or a sharp rise in population that poses the worst threat to the land, but an attitude that does not know its limits and shamelessly and brutally milks the resources of the earth to the point of their depletion.[19] Perhaps the only way that we will ever learn to respect the environment that we have farmed, fished, and mined to death is to accept the challenge of welcoming another onto our own territory. Only when we have allowed the stranger to teach us about our own limits will we develop a healthy attitude towards the land, the environment, and its natural resources. Thus, far from causing a depletion of our natural resources, the welcoming of the other onto our own territory, the willingness to learn to share with this other and appreciate our limits, can in fact result in a more ethical and sustainable approach to our resources, thereby increasing the productivity of the land rather than decreasing it.

This brings us to our third objection. While the other is clearly not a threat to our resources, the problem of the threat posed to our identity by the immigrant remains. It is a common fear that the influx of immigrants poses a threat to national identity, that is to say, to our culture, values, religion, etc. We believe that the influx of immigrants, with their different values and lifestyles, poses a threat to our culture. Thus, in order to protect our identity from the surge of otherness, we often choose a stance of protectionism. This attitude is evident today in many Western countries that believe they are engaged in a "clash of civilizations",[20] between Western and non-Western values that are often depicted as "primitive", "archaic", and even "immoral". The question that arises with great anxiety is whether Western values can withstand the flux of "obscurantism" flowing through the borders of Western nations or whether we are not witnessing today a new "barbarian invasion". Our attitude when faced with this threat has been to push back on immigration under the pretext that it is deeply

---

[18]    This is incidentally the connection that Ecofeminism makes between the uncurbed power of patriarchy over women and the degradation of the environment. Ecofeminists such as Vandana Shiva have noted the following: "Wherever women acted against ecological destruction or/and the threat of atomic annihilation, they immediately became aware of the connection between patriarchal violence against women, other people and nature and that: In defying this patriarchy we are loyal to future generations and to life and this planet itself" (Shiva and Mies 1993, p. 14).

[19]    This myth that overpopulation is the cause behind the degradation of the environment has been debunked in a number of articles such as this one from the World Watch Institute, which states that "[d]espite rising consumption in the developing world, industrial countries remain responsible for the bulk of the world's resource consumption—as well as the associated global environmental degradation. Yet there is little evidence that the consumption locomotive is braking, even in the United States, where most people are amply supplied with the goods and services needed to lead a dignified life" (World Watch Institute, "The State of Consumption Today", 21 November 2018). Charles Eisenstein follows up on these observations by arguing that "this means that overpopulation is a red herring. Of course, at some point, preferably soon, population growth must end, but overpopulation is a diversion from more fundamental issues. Lurking behind the spectre of population growth lies a more challenging problem: economic growth" (Charles Eisenstein, The Guardian, 28 March 2014).

[20]    For more on this notion cf. (Huntington 1993).

detrimental and threatening to our culture and identity as Westerners. The safeguarding of culture thus depends on a systematic rejection of the others at our door.

Again, our text reverses the situation. According to our text, culture actually arises only when identity is transcended towards otherness. It is only when Cain is exiled from his home, his territory, might I say, from his identity, that he births the generations that would give rise to culture and civilization. It is only when he is cast east of Eden that he engenders the three founders of civilization—Jabal the tent-dweller, Jubal the musician, and Tubal-cain the forger of instruments—and builds the first city (Gen. 4: 20–22). Thus, the curse of his expulsion from the land of Eden towards the other is really a blessing in disguise. It is only when Cain leaves his Eden, his *terroir*, his identity that he gives rise to civilization and culture. Thus, in our text, culture and civilization is birthed out of a journey *out* of identity rather than from a recoiling to identitarian attitudes. However, more needs to be said on this. What is it about the overcoming of identity that is so essential to the birth of culture? For this, we need to further understand what identity really means.

Identity comes from the Latin *idem*, which means "the same". Thus, the identity of a given community constitutes that which unites its members under the banner of sameness—that which constitutes the *common* values, way of life, food, dress, and religion of its members. I am reminded here of the enterprise of Babel, which, essentially, could be read as a primitive form of identity politics, that is, the creation of a community entirely defined by what unites its members, what makes them the same over and against what separates and differentiates them. We remember of course that God was hardly pleased with this endeavor. The Hebrew Bible thus has typically viewed identity with suspicion. The question, of course, is why. The main reason is that identity, with its emphasis on the same, is essentially allergic to otherness. The problem with this is that, for the Biblical ethos, identity without otherness is sterile. Only upon encountering its other is a given identity fecund. Fecundity, which in turn gives rise to the creations at the source of culture, is possible only through the encounter between identity and difference. It is only when identity meets difference that the fecund creations of culture and civilization are birthed.

So, it is only when Adam and Eve leave the garden towards a new and other place that they birth their first sons. Only when Cain leaves behind his home and land towards the great unknown does he begin to birth civilization. Only when the identitarian endeavor of Babel is interrupted do diverse and rich cultures emerge. Finally, only when Abraham leaves his father's house does he receive the blessing of fecundity. Thus, in the Biblical sense, culture is always the product of an encounter with otherness. Culture and civilization emerge thus dialogically, inter-subjectively rather than in the mind of a single individual or community.[21] The idea that civilization was born from a single people called to serve as the "light unto the nations" speaks to a profound misunderstanding of how human creativity emerges. We now know that the Enlightenment, hereto thought to have emerged ex nihilo in the West, was, in fact, the product of centuries of an inter-mingling between Greek, Arab, Hebrew, and Western cultures. Culture is born and revitalized by a fruitful encounter between identity and difference. Thus, far from protecting our culture by isolating ourselves from what is perceived as a dangerous intrusion of migrants, we are in fact accelerating its demise. Culture and civilization are born and reborn from

---

21　This idea has been developed by Senghor in his concept of "Civilization of the Universal". Quoting Teillhard de Chardin, Senghor observes: "Teilhard shows us that conflicts between human groups—technico-professional groups or 'classes,' nations, races—are natural facts; moreover, that they are necessary steps in the process of socialization . . . that a movement of pan-human convergence has been set in motion by the very tension and the power of our technical means—peaceful and military. From this movement, the planetary civilization will emerge, a symbiosis of all particular civilizations; and the scientist invites us, the underdeveloped people, to help construct the Civilization of the Universal" (Senghor 1964, p. 139). Senghor goes on to quote Teilhard de Chardin directly as he describes the birth of civilization as a fecundation between different worldviews and practices: "Before the last upheavals that awakened the earth, peoples were only superficially alive; a world of energies was still asleep within each of them. Well, I imagine that these powers, still dormant within each natural human unity, in Europe, Asia, everywhere, are stirring and trying to come to light at this very moment; not to oppose and devour one another, but to rejoin and interfecundate one another. Fully conscious nations are needed for a total earth" (Teilhard de Chardin, *Oeuvres III*, quoted in Senghor 1964, pp. 139–40).

the ongoing fruitful contact between the same and the other. To isolate oneself is to die as a culture and as a civilization.

This however leads to a profound realization that ethics is at the foundation of civilization.[22] It is not our reason and its technical and scientific prowess that have led us to where we are as a civilization but, rather, our ability to engage with difference. What has made America great is not merely its "white Anglo-Saxon protestant ethic", but rather its melting-pot character where American civilization found itself constantly rebirthed, rejuvenated, and dynamized by a continual influx of otherness. It is the American ethics of inclusion and its hospitable stance to migrants that has made its strength. Thus, the ethical and welcoming stance towards the strangers in our midst actually does not threaten culture but, rather, revitalizes it. It is really the loss of this ethical stance that poses the deepest threat to our culture and to our values. Culture and civilization are grounded in ethics, in a welcoming stance towards humanity. To forget this sense of hospitality is to spell out the demise of the very culture we are trying to safeguard. We see this in our story: although Jabal, Jubal, and Tubal-cain have all three birthed culture through their life-giving creativity, their father, Lamech, is a violent man. His violence is seventy-seven-fold that of Cain's (Gen. 4: 23–24). It is this violence that will eventually spell the demise of all of the creations of the sons in the terrible and fatal flood that would annihilate everything (Gen. 6). Thus, our story teaches that where culture and civilization lack ethics, they eventually die. Let us remain keenly aware of the warnings in this text, as violence is on the rise in our Western nations in the name of culture! To lose a sense of our humanity, of the ethics of hospitality, is to eventually destroy ourselves and the civilization that we have created.

The Story of Cain and Abel is much more than a mere tale of two brothers buried in an ancient and forgotten past. It is a tale for our times, a mirror placed in front of us and a warning of where our present paranoia towards the stranger might lead. For too long, we have set ourselves up as an example of enlightenment and moral purity. We have become the judges and enforcers of morality in the world, with little awareness of our profound moral depravity. Like Cain, we have thought ourselves at the center of the world, with everyone else relegated to "second-world" or "third-world" status, with little awareness of the inevitable violent and murderous possibilities hidden within that stance. Like Cain, we have taken moral initiatives and offered sacrifices to the past gods of colonialism and imperialism and to today's gods of nation-building and just war; we have thought of ourselves somehow to be "chosen", special, privileged. Like Cain, we have been oblivious to our profound lack of awareness of the other, the stranger, and to our own calloused, inhuman attitude towards the ones that begged us for food, shelter, and safety. We have made the other to be the threat when, in fact, we are the ones that pose the greatest danger, not only to ourselves as a civilization, but also to our common humanity. Perhaps it is time for us to go back to this ancient text, pore over its pages, and reflect on who we are today and who we might eventually become if we choose to continue in the present trajectory of entitlement, xenophobia, and isolationism.

**Funding:** This research received no external funding.

**Conflicts of Interest:** The author declares no conflict of interest.

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
