# Peer review of "Cain and Abel: Re-Imagining the Immigration ‘Crisis’"

_religions, doi:10.3390/rel11030112_

Round 1

Reviewer 1 Report

This article is original, much appropriate and relevant.

(check spelling in line 190)

Author Response

Dear reviewer, 

Thank you for your positive feedback!

Reviewer 2 Report

The article is an interesting midrashic--interpretive understanding--of the Cain & Abel story/conflict, suggesting that the "root" of this story is a conflict between the sedentary and the nomadic.  Intriguing.

However, the author needs to stress that the ultimate outcome of this othering results not only in violence but murder, and that this life-taking not only colors but frames the entire narrative.

Without doing so, the author, at least according to this reviewer, somewhat demeans his/her overall argument.  While othering may not always result in violence and/or murder and/or genocide, this first example of murder sets a stage for the subsequent course of human history.

Author Response

The reviewer made the following important point: "However, the author needs to stress that the ultimate outcome of this othering results not only in violence but murder, and that this life-taking not only colors but frames the entire narrative."

My reponse: Thank you for this observation. I have attempted to make the connection clearer between the conceptual violence inherent in the rejection/demonization of the exiled and its ultimate consequences in extermination and murder. To do so, I have added a whole paragraph developing this connection, adding examples of how both the Nazi and Rwandian genocides stemmed directly from the process of demonization and dehumanization of its victims. I then proceeded to problematize the current demonizations of immigrants (more specifically Mexican immigrants) by the current administration in light of this. 

Thank you again for your feedback and encouraging words!

Round 2

Reviewer 2 Report

Authors have addressed this reviewer's concerns. Paper is now worthy of publication.

Author Response

(The authors gave the same response as above.)
